# Provably Feedback-Efficient Reinforcement Learning via Active Reward Learning

**Dingwen Kong**
School of Mathematical Sciences
Peking University
dingwenk@pku.edu.cn

**Lin F. Yang**
Department of Electrical and Computer Engineering
University of California, Los Angeles
linyang@ee.ucla.edu

## Abstract

An appropriate reward function is of paramount importance in specifying a task in reinforcement learning (RL). Yet, it is known to be extremely challenging in practice to design a correct reward function for even simple tasks. Human-in-the-loop (HiL) RL allows humans to communicate complex goals to the RL agent by providing various types of feedback. However, despite achieving great empirical successes, HiL RL usually requires *too much* feedback from a human teacher and also suffers from insufficient theoretical understanding. In this paper, we focus on addressing this issue from a theoretical perspective, aiming to provide provably feedback-efficient algorithmic frameworks that take human-in-the-loop to specify rewards of given tasks. We provide an *active-learning*-based RL algorithm that first explores the environment without specifying a reward function and then asks a human teacher for only a few queries about the rewards of a task at some state-action pairs. After that, the algorithm guarantees to provide a nearly optimal policy for the task with high probability. We show that, even with the presence of random noise in the feedback, the algorithm only takes $\widetilde{O}(H\dim_R^2)$ queries on the reward function to provide an $\varepsilon$-optimal policy for any $\varepsilon > 0$. Here $H$ is the horizon of the RL environment, and $\dim_R$ specifies the complexity of the function class representing the reward function. In contrast, standard RL algorithms require to query the reward function for at least $\Omega(\mathrm{poly}(d, 1/\varepsilon))$ state-action pairs where $d$ depends on the complexity of the environmental transition.

## 1 Introduction

A suitable reward function is essential for specifying a reinforcement learning (RL) agent to perform a complex task. Yet obvious approaches such as hand-designed reward is not scalable for large number of tasks, especially in the multitask settings [Wilson et al., 2007, Brunskill and Li, 2013, Yu et al., 2020, Sodhani et al., 2021], and can also be extremely challenging (e.g., [Ng et al., 1999, Marthi, 2007] shows even intuitive reward shaping can lead to undesired side effects). Recently, a popular framework called Human-in-the-loop (HiL) RL [Knox and Stone, 2009, Christiano et al., 2017, MacGlashan et al., 2017, Ibarz et al., 2018, Lee et al., 2021, Wang et al., 2022] gains more interests as it allows humans to communicate complex goals to the RL agent directly by providing various types of feedback. In this sense, a reward function can be learned automatically and can also be corrected at proper times if unwanted behavior is happening. Despite its promising empirical performance, HiL algorithms still suffer from insufficient theoretical understanding and possess drawbacks Arakawa et al. [2018], e.g., it assumes humans can give precise numerical rewards and do so without delay and at every time step, which are usually not true. Moreover, these approaches usually train on every new task separately and cannot incorporate exiting experiences.

36th Conference on Neural Information Processing Systems (NeurIPS 2022).

In this paper, we attempt to address the above issues of incorporating humans' feedback in RL from a theoretical perspective. In particular, we would like to address (1) the high *feedback complexity* issue – i.e., the algorithms in practice usually require large amount feedback from humans to be accurate; (2) feedback from humans can be noisy and non-numerical; (3) in need for support of multiple tasks. In particular we consider a fixed unknown RL environment, and formulate a task as an unknown but fixed reward function. A human who wants the agent to accomplish the task needs to communicate the reward to the agent. It is not possible to directly specify the parameters of the reward function as the human may not know it exactly as well, but is able to specify good actions at any given state. To capture the non-numerical feedback issue, we assume that the feedback we can get for an action is only binary – whether an action is "good" or "bad". We further assume that the feedback is noisy in the sense that the feedback is only correct with certain probability. Lastly, we require that the algorithm, after some initial exploration phase, should be able to accomplish multiple tasks by only querying the reward rather than the environment again.

In the supervised learning setting, if we only aim to learn a reward function, the feedback complexity can be well-addressed by the active learning framework Settles [2009], Hanneke et al. [2014] – an algorithm only queries a few samples of the reward entries and then provide a good estimator. Yet this become challenging in the RL setting as it is a sequential decision making problem – state-action pairs that are important in the supervised learning setting may not be accessible in the RL setting. Therefore, to apply similar ideas in RL, we need a way to explore the environment and collect samples that are important for reward learning. Fortunately, there were a number of recent works focusing "reward-free" exploration Jin et al. [2020a], Wang et al. [2020a] on the environment. Hence, applying such an algorithm would not affect the feedback complexity. Additionally it is possible for us to reuse the collected data for multiple tasks.

Our proposed theoretical framework is a non-trivial integration of reward-free reinforcement learning and active learning. The algorithm possesses two phases: in phase I, it performs reward-free RL to explore the environment and collect the small but necessary amount of the information about the environment; in phase II, the algorithm performs active learning to query the human for the reward at only a few state-action pairs and then provide a near optimal policy for the tasks with high probability. The algorithm is guaranteed to work even the feedback is noisy and binary and can solve multiple tasks in phase II. Below we summarize our contributions:

1. We propose a theoretical framework for incorporating humans' feedback in RL. The framework contains two phases: an unsupervised exploration and an active reward learning phase. Since the two phases are separated, our framework is suitable for multi-task RL.

2. Our framework deals with a general and realistic case where the human feedback is stochastic and binary-i.e., we only ask the human teacher to specify whether an action is "good" or "bad". We design an efficient active learning algorithm for learning the reward function from this kind of feedback.

3. Our query complexity is minimal because it is independent of both the environmental complexity $d$ and target policy accuracy $\varepsilon$. In contrast, standard RL algorithms require query the reward function for at least $\Omega(\text{poly}(d, 1/\varepsilon))$ state-action pairs. Thus our work provides a theoretical validation for the recent empirical HiL RL works where the number of queries is significantly smaller than the number of environmental steps.

4. Moreover, we shows the efficacy of our framework in the offline RL setting, where the environmental transition dataset is given beforehand.

## 1.1 Related Work

**Sample Complexity of Tabular and Linear MDP.** There is a long line of theoretical work on the sample complexity and regret bound for tabular MDP. See, e.g., [Kearns and Singh, 2002, Jaksch et al., 2010, Azar et al., 2017, Jin et al., 2018, Zanette and Brunskill, 2019, Agarwal et al., 2020b, Wang et al., 2020b, Li et al., 2022]. The linear MDP is first studied in Yang and Wang [2019]. See, e.g., [Yang and Wang, 2020, Jin et al., 2020b, Zanette et al., 2020a, Ayoub et al., 2020, Zhou et al., 2021a,b] for sample complexity and regret bound for linear MDP.

**Unsupervised Exploration for RL.** The reward-free exploration setting is first studied in Jin et al. [2020a]. This setting is later studied under different function approximation scheme: tabular [Kauf-

mann et al., 2021, Ménard et al., 2021, Wu et al., 2022], linear function approximation [Wang et al., 2020a, Zanette et al., 2020b, Zhang et al., 2021, Huang et al., 2022, Wagenmaker et al., 2022, Agarwal et al., 2020a, Modi et al., 2021], and general function approximation [Qiu et al., 2021, Kong et al., 2021, Chen et al., 2022]. Besides, Zhang et al. [2020], Yin and Wang [2021] study task-agnostic RL, which is a variety of reward-free RL. Wu et al. [2021] studies multi-objective RL in the reward-free setting. Bai and Jin [2020], Liu et al. [2021] studies reward-free exploration in Markov games.

**Active Learning.** Active learning is relatively well-studied in the context of unsupervised learning. See, e.g., Dasgupta et al. [2007], Balcan et al. [2009], Settles [2009], Hanneke et al. [2014] and the references therein. Our active reward learning algorithm is inspired by a line of works [Cesa-Bianchi et al., 2009, Dekel et al., 2010, Agarwal, 2013] considering online classification problem where they assume the response model $P(y|x)$ is linear parameterized. However, their works can not directly apply to the RL setting and also the non-linear case. There are also many empirical study-focused paper on active reward learning. See, e.g., [Daniel et al., 2015, Christiano et al., 2017, Sadigh et al., 2017, Bıyık et al., 2019, 2020, Wilde et al., 2020, Lindner et al., 2021, Lee et al., 2021]. Many of them share similar algorithmic components with ours, like information gain-based active query and unsupervised pre-training. But they do not provide finite query complexity bounds.

# 2 Preliminaries

## 2.1 Episodic Markov Decision Process

In this paper, we consider the finite-horizon Markov decision process (MDP) $M = (\mathcal{S}, \mathcal{A}, P, r, H, s_1)$, where $\mathcal{S}$ is the state space, $\mathcal{A}$ is the action space, $P = \{P_h\}_{h=1}^H$ where $P_h : \mathcal{S} \times \mathcal{A} \to \triangle(\mathcal{S})$ are the transition operators, $r = \{r_h\}_{h=1}^H$ where $r_h : \mathcal{S} \times \mathcal{A} \to \{0, 1\}$ are the deterministic *binary* reward functions, and $H$ is the planning horizon. Without loss of generality, we assume that the initial state $s_1$ is fixed.[1] In RL, an agent interacts with the environment episodically. Each episode consists of $H$ time steps. A deterministic policy $\pi$ chooses an action $a \in \mathcal{A}$ based on the current state $s \in \mathcal{S}$ at each time step $h \in [H]$. Formally, $\pi = \{\pi_h\}_{h=1}^H$ where for each $h \in [H]$, $\pi_h : \mathcal{S} \to \mathcal{A}$ maps a given state to an action. In each episode, the policy $\pi$ induces a trajectory $s_1, a_1, r_1, s_2, a_2, r_2, ..., s_H, a_H, r_H, s_{H+1}$ where $s_1$ is fixed, $a_1 = \pi_1(s_1)$, $r_1 = r_1(s_1, a_1)$, $s_2 \sim P_1(\cdot|s_1, a_1)$, $a_2 = \pi_2(s_2)$, etc.

We use Q-function and V-function to evaluate the long-term expected cumulative reward in terms of the current state (state-action pair), and the policy deployed. Concretely, the Q-function and V-function are defined as: $Q_h^\pi(s, a) = \mathbb{E}\big[\sum_{h'=h}^H r_{h'}(s_{h'}, a_{h'})|s_h = s, a_h = a, \pi\big]$ and $V_h^\pi(s) = \mathbb{E}\big[\sum_{h'=h}^H r_{h'}(s_{h'}, a_{h'})|s_h = s, \pi\big]$. We denote the optimal policy as $\pi^* = \{\pi_h^*\}_{h\in[H]}$, optimal values as $Q_h^*(s, a)$ and $V_h^*(s)$. Sometimes it is convenient to consider the Q-function and V-function where the true reward function is replaced by a estimated one $\hat{r} = \{\hat{r}_h\}_{h\in[H]}$. We denote them as $Q_h^\pi(s, a, \hat{r})$ and $V_h^\pi(s, \hat{r})$. We also denote the corresponding optimal policy and value as $\pi^*(\hat{r})$, $Q_h^*(s, a, \hat{r})$ and $V_h^*(s, \hat{r})$.

**Additional Notations.** We define the infinity-norm of function $f : \mathcal{S} \times \mathcal{A} \to \mathbb{R}$ as $\|f\|_\infty = \sup_{(s,a)\in\mathcal{S}\times\mathcal{A}} |f(s, a)|$. For a set of state-action pairs $\mathcal{Z} \subseteq \mathcal{S} \times \mathcal{A}$ and a function $f : \mathcal{S} \times \mathcal{A} \to \mathbb{R}$, we define $\|f\|_{\mathcal{Z}} = \left(\sum_{(s,a)\in\mathcal{Z}} f(s, a)^2\right)^{1/2}$.

# 3 Technical Overview

In this section we give a overview of our learning scenario and notations, as well as the main techniques. The learning process divides into two phases.

## 3.1 Phase 1: Unsupervised Exploration

The first step is to explore the environment without reward signals. Then we can query the human teacher about the reward function in the explored region. We adopt the *reward-free exploration* technique developed in Jin et al. [2020a], Wang et al. [2020a]. The agent is encouraged to do

---

[1]For a general initial distribution $\rho$, we can treat it as the first stage transition probability, $P_1$.

exploration by maximizing the cumulative *exploration bonus*. Concretely, we gather $K$ trajectories $\mathcal{D} = \{(s_h^k, a_h^k)\}_{(h,k)\in[H]\times[K]}$ by interacting with the environment. We can strategically choose which policy to use. At the beginning of the $k$-th episode, we calculate a policy $\pi_k$ based on the history of the first $k-1$ episodes and use $\pi_k$ to induce a trajectory $\{(s_h^k, a_h^k)\}_{h\in[H]}$.

A similar approach called *unsupervised pre-training* [Sharma et al., 2020, Liu and Abbeel, 2021] has been successfully used in practice. Concretely, in unsupervised pre-training agents are encouraged to do exploration by maximizing various *intrinsic rewards*, such as prediction errors [Houthooft et al., 2016] and count-based state-novelty [Tang et al., 2017].

## 3.2 Phase 2: Active Reward Learning

The second step is to learn a proper reward function from human feedback. Our work assumes that the underlying valid reward is 1-0 binary, which is interpreted as good action and bad action. We remark that RL problems with binary rewards represent a large group of RL problems that are suitable and relatively easy for having human-in-the-loop. A representative group of problems is the binary judgments: For example, suppose we want a robot to learn to do a backflip. A human teacher will judge whether a flip is successful and assign a reward of 1 for success and 0 for failure. Furthermore, our framework can also be generalized to RL problems with $n$-uniform discrete rewards. The detailed discussion is defered to Appendix E.2 due to space limit.

Concretely, consider a fixed stage $h \in [H]$, and we are trying to learn $r_h$ from the human response. Each time we can query a datum $z = (s, a)$ and receive an independent random response $Y \in \{0, 1\}$ from the human expert, with distribution:

$$P(Y = 1|z) = 1 - P(Y = 0|z) = f_h^*(z).$$

Here $f_h^*$ is the human response model and needs to be learned from data. We assume that the underlying valid reward of $z$ can be determined by $f_h^*(z)$ in the following manner:

$$r_h(z) = \begin{cases} 1, & f_h^*(z) > 1/2 \\ 0, & f_h^*(z) \le 1/2. \end{cases}$$

Note that the query returns 1 with a probability greater than $\frac{1}{2}$ if and only if the underlying valid reward is 1. To make the number of queries as small as possible, we choose a small subset of informative data to query the human. We adopt ideas in the *pool-based active learning* literature and select informative queries greedily. We show that only $\widetilde{O}(H\dim_R^2)$ queries need to be answered by the human teacher. The active query method is widely used in human-involved reward learning in practice and shows superior performance than uniform sampling Christiano et al. [2017], Ibarz et al. [2018], Lee et al. [2021].

After we learn a proper reward function $\hat{r}$, we use past experience $\mathcal{D}$ and $\hat{r}$ to plan for a good policy. Note that in this phase we are not allowed for further interaction with the environment. In the multi-task RL setting, we can run Phase 2 for multiple times and reuse the data collected in Phase 1.

Now we discuss the efficacy of our framework. A naive approach for reward learning via human feedback is asking the human teacher to evaluate the reward function in each round. This approach results in equal environmental steps and number of queries. This high query frequency is unacceptable for large-scale problems. For example, in Lee et al. [2021] the agent learns complex tasks with very few queries ($\sim 10^2$ to $10^3$ queries) to the human compared to the number of environmental steps ($\sim 10^6$ steps) by utilizing active query technique. From the theoretical perspective, usual RL sample complexity bound scales with $\propto \mathrm{poly}(d, \frac{1}{\varepsilon})$, where $d$ is the complexity measure of the environmental transition and $\varepsilon$ is the target policy accuracy. This quantity can be huge when the environment is complex (i.e., $d$ is large) or with small target accuracy. Our query complexity is desirable since it is independent of both $d$ and $1/\varepsilon$.

## 4 Pool-Based Active Reward Learning

In this section we formally introduce our algorithm for active reward learning. We consider a fixed stage $h$ and learn $r_h$ by querying a small subset of $\mathcal{Z}_h = \{(s_h^k, a_h^k)\}_{k\in[K]}$. We omit the subscript $h$ in this section, i.e., we use $\mathcal{Z}, z_k, r, f^*$ to denote $\mathcal{Z}_h, z_h^k, r_h, f_h^*$ in this section. Since $\mathcal{Z}$ is given

before the learning process starts, we refer to this learning scenario as *pool-based* active learning. Our purpose is to learn a reward function $\hat{r}(\cdot)$ such that $r(z) = \hat{r}(z)$ for most of $z$ in $\mathcal{Z}$. At the same time, we hope the number of queries can be as small as possible.

We assume $\mathcal{F}$ is a pre-specified function class to learn $f^*$ from, and $\mathcal{F}$ is known as a prior. We assume that $\mathcal{F}$ has enough expressive power to represent the human response. Concretely, we assume the following *realizability*.

**Assumption 1** (Realizability). $f^* \in \mathcal{F}$.

The learning problem can be arbitrarily difficult, especially when $f^*(z)$ is close to $\frac{1}{2}$, in which case it will be difficult to determine the true value of $r(z)$. To give a problem-dependent bound, we assume the following *bounded noise* assumption. In the literature on statistical learning, this assumption is also referred to as *Massart noise* [Massart and Nédélec, 2006, Giné and Koltchinskii, 2006, Hanneke et al., 2014]. Our framework can also work under the *low noise* assumption - due to space limit, we defer the discussion to Appendix E.3.

**Assumption 2** (Bounded Noise). *There exists $\Delta > 0$, such that for all $z \in \mathcal{S} \times \mathcal{A}$, $|f^*(z) - \frac{1}{2}| > \Delta$.*

The value of the margin $\Delta$ depends on the intrinsic difficulty of the reward learning problem and the capacity of the human teacher. For example, if the reward is rather easy to specify and the human teacher is a field expert, and can always give the right answer with a probability of at least 80%, then $\Delta$ will be 0.3. But if the learning problem is hard or the human teacher is unfamiliar with the problem and can only give near-random answers, then $\Delta$ will be very small. But in that case, we won't hope the human teacher can help us in the first place. So a typical good value for $\Delta$ should be a constant.

**Examples**  We give two examples of $\mathcal{F}$ that is frequently studied in the active learning literature. In the linear model, the function class $\mathcal{F}$ consists of $f$ in the following form: $f(z) = \frac{\langle \phi(z), w \rangle + 1}{2}$. In the logistic model, the function class $\mathcal{F}$ consists of $f$ in the following form: $f(z) = \frac{\exp \langle \phi(z), w \rangle}{1 + \exp \langle \phi(z), w \rangle}$. Here $\phi : \mathcal{S} \times \mathcal{A} \to \mathbb{R}^d$ is a fixed and known feature extractor, and $w \in \mathbb{R}^d$.

The complexity of $\mathcal{F}$ essentially depends on the learning complexity of the human response model. We use the following *Eluder dimension* [Russo and Van Roy, 2014] to characterize the complexity of $\mathcal{F}$. The eluder dimension serves as a common complexity measure of a general non-linear function class in both reinforcement learning literature [Osband and Van Roy, 2014, Ayoub et al., 2020, Wang et al., 2020c, Jin et al., 2021a] and active learning literature [Chen et al., 2021].

**Definition 1** (Eluder Dimension). *Let $\varepsilon \geq 0$ and $\mathcal{Z} = \{(s_i, a_i)\}_{i=1}^n \subseteq \mathcal{S} \times \mathcal{A}$ be a sequence of state-action pairs.*
*(1) A state-action pair $(s, a) \in \mathcal{S} \times \mathcal{A}$ is $\varepsilon$-dependent on $\mathcal{Z}$ with respect to $\mathcal{F}$ if any $f, f' \in \mathcal{F}$ satisfying $\|f - f'\|_{\mathcal{Z}} \leq \varepsilon$ also satisfies $|f(s, a) - f'(s, a)| \leq \varepsilon$.*
*(2) An $(s, a)$ is $\varepsilon$-independent of $\mathcal{Z}$ with respect to $\mathcal{F}$ if $(s, a)$ is not $\varepsilon$-dependent on $\mathcal{Z}$.*
*(3) The $\varepsilon$-eluder dimension $\dim_E(\mathcal{F}, \varepsilon)$ of a function class $\mathcal{F}$ is the length of the longest sequence of elements in $\mathcal{S} \times \mathcal{A}$ such that, for some $\varepsilon' \geq \varepsilon$, every element is $\varepsilon'$-independent of its predecessors.*
*(4) The eluder dimension of a function class $\mathcal{F}$ is defined as $\dim_E(\mathcal{F}) := \limsup_{\alpha \downarrow 0} \frac{\dim_E(\mathcal{F}, \alpha)}{\log(1/\alpha)}$.*

We remark that a wide range of function classes, including linear functions, generalized linear functions and bounded degree polynomials, have bounded eluder dimension.

**Definition 2** (Covering Number and Kolmogorov Dimension). *For any $\varepsilon > 0$, there exists an $\varepsilon$-cover $\mathcal{C}(\mathcal{F}, \varepsilon) \subseteq \mathcal{F}$ with size $|\mathcal{C}(\mathcal{F}, \varepsilon)| \leq \mathcal{N}(\mathcal{F}, \varepsilon)$, such that for any $f \in \mathcal{F}$, there exists $f' \in \mathcal{C}(\mathcal{F}, \varepsilon)$ with $\|f - f'\|_\infty \leq \varepsilon$. The Kolmogorov dimension of $\mathcal{F}$ is defined as: $\dim_K(\mathcal{F}) := \limsup_{\alpha \downarrow 0} \frac{\log(\mathcal{N}(\mathcal{F}, \alpha))}{\log(1/\alpha)}$.*

The Kolmogorov dimension is also bounded by $O(d)$ for linear/generalized linear function class. Throughout this paper, we denote $\dim(\mathcal{F}) := \max\{\dim_E(\mathcal{F}), \dim_K(\mathcal{F})\}$ as the complexity measure of $\mathcal{F}$. When $\mathcal{F}$ is the class of $d$-dimensional linear/generalized linear functions, $\dim(\mathcal{F})$ is bounded by $O(d)$.

## 4.1 Algorithm

We describe our algorithm for learning the human response model and the underlying reward function. We sequentially choose which data points to query. Denote $\mathcal{Z}_k$ the first $k$ points that we decide to

query and initial $\mathcal{Z}_0$ to be an empty set. For each $z \in \mathcal{Z}$, we use the following bonus function to measure the information gain of querying $z$, i.e., how much new information $z$ contains compared to $\mathcal{Z}_{k-1}$:

$$b_k(\cdot) \leftarrow \sup_{f,f' \in \mathcal{F}, \|f-f'\|_{\mathcal{Z}_{k-1}} \leq \beta} |f(\cdot) - f'(\cdot)|.$$

We then simply choose $z_k$ to be $\arg\max_{z \in \mathcal{Z}} b_k(z)$. After the $N$ query points are determined, we query their labels from a human. The human response model is then learned by solving a least-squares regression:

$$\widetilde{f} \leftarrow \min_{f \in \mathcal{F}} \sum_{z \in \mathcal{Z}_N} (f(z) - l(z))^2.$$

The human response model is used for estimating the underlying reward function. We round $\widetilde{f}$ to the cover $\mathcal{C}(\mathcal{F}, \Delta/2)$ to ensure that there are a finite number of possibilities of such functions – this gives us the convenience of applying union bound in our analysis. Indeed, we believe a more refined analysis would remove the requirement of rounding but will make the analysis much more involved. The whole algorithm is presented in Algorithm 1. Note that such an interactive mode with the human teacher is *non-adaptive* since all queries are given to the human teacher in one batch. This property makes our algorithm desirable in practice. Here we assume the value of $\Delta$ is known as a prior. We can extend our results to the case where $\Delta$ is unknown. Due to space limit, we defer the discussion to Appendix E.1.

---

**Algorithm 1** Active Reward Learning($\mathcal{Z}, \Delta, \delta$)

---

**Input:** Data Pool $\mathcal{Z} = \{z_i\}_{i \in [T]}$, margin $\Delta$, failure probability $\delta \in (0, 1)$
$\mathcal{Z}_0 \leftarrow \{\}$ //Query Dataset
Set $N \leftarrow C_1 \cdot \frac{(\dim^2(\mathcal{F}) + \dim(\mathcal{F}) \cdot \log(1/\delta)) \cdot (\log^2(\dim(\mathcal{F})))}{\Delta^2}$
**for** $k = 1, 2, ..., N$ **do**
   $\beta \leftarrow C_2 \cdot \sqrt{\log(1/\delta) + \log N \cdot \dim(\mathcal{F})}$
   Set the bonus function: $b_k(\cdot) \leftarrow \sup_{f,f' \in \mathcal{F}, \|f-f'\|_{\mathcal{Z}_{k-1}} \leq \beta} |f(\cdot) - f'(\cdot)|$
   $z_k \leftarrow \arg\max_{z \in \mathcal{Z}} b_k(z)$
   $\mathcal{Z}_k \leftarrow \mathcal{Z}_{k-1} \cup \{z_k\}$
**end for**
**for** $z \in \mathcal{Z}_N$ **do**
   Ask the human expert for a label $l(z) \in \{0, 1\}$
**end for**
Estimate the human model as $\widetilde{f} = \arg\min_{f \in \mathcal{F}} \sum_{z \in \mathcal{Z}_N} (f(z) - l(z))^2$
Let $\hat{f} \in \mathcal{C}(\mathcal{F}, \Delta/2)$ such that $\|\hat{f} - \widetilde{f}\|_\infty \leq \Delta/2$
Estimate the underlying true reward: $\hat{r}(\cdot) = \begin{cases} 1, & \hat{f}(\cdot) > 1/2 \\ 0, & \hat{f}(\cdot) \leq 1/2 \end{cases}$
**return:** The estimated reward function $\hat{r}$.

---

### 4.2 Theoretical Guarantee

**Theorem 1.** *With probability at least $1 - \delta$, for all $z \in \mathcal{Z}$, we have, $\hat{r}(z) = r(z)$. The total number of queries is bounded by* $O\left(\frac{(\dim^2(\mathcal{F}) + \dim(\mathcal{F}) \cdot \log(1/\delta)) \cdot (\log^2(\dim(\mathcal{F})))}{\Delta^2}\right)$.

**Proof Sketch** The first step is to show that the sum of bonus functions $\sum_{k=1}^K b_k(z_k)$ is bounded by $O(d\sqrt{K})$ using ideas in Russo and Van Roy [2014]. Note that the bonus function $b_k(\cdot)$ is non-increasing. Thus we can show that after selecting $N = \widetilde{O}(\frac{d^2}{\Delta^2})$ points, for all $z \in \mathcal{Z}$, the bonus function of $z$ does not exceed $\Delta$. By the bounded-noise assumption, we know that the reward label for $z$ is correct for all $z$.

## 5 Online RL with Active Reward Learning

In this section we consider how to apply active reward learning method in the online RL setting. In this setting the agent is allowed to actively explore the environment without reward signal in the exploration phase. We consider both tabular MDP and linear MDP cases.

## 5.1 Linear MDP with Positive Features

The linear MDP assumption was first studied in Yang and Wang [2019] and then applied in the online setting Jin et al. [2020b]. It is assumed that the agent is given a feature extractor $\phi : \mathcal{S} \times \mathcal{A} \to \mathbb{R}^d$ and the transition model can be predicted by linear functions of the give feature extractor. In our work, we additionally assume that the coordinates of $\phi$ are all *positive*. This assumption essentially reduce the linear MDP model to the soft state aggregation model [Singh et al., 1994, Duan et al., 2019]. As will be seen later, the latent state structure helps the learned reward function to generalize.

**Assumption 3** (Linear MDP with Non-Negative Features). *For all $h \in [H]$, we assume that there exists a function $\mu_h : \mathcal{S} \to \mathbb{R}^d$ such that $P_h(s'|s, a) = \langle \mu_h(s'), \phi(s, a) \rangle$. Moreover, for all $(s, a) \in \mathcal{S} \times \mathcal{A}$, the coordinates of $\phi(s, a)$ and $\mu_h(s, a)$ are all non-negative.*

## 5.2 Exploration Phase

In the exploration phase, inspired by former works on reward-free RL, we use optimistic least-squares value iteration (LSVI) based algorithm with zero reward. In the linear case, for any $V : \mathcal{S} \to \mathbb{R}$, we estimate $P_h V$ in the following manner

$$\widehat{P}_h^k V(\cdot, \cdot) \leftarrow w^T \phi(\cdot, \cdot), \text{ where } w \leftarrow \arg\min_{w \in \mathbb{R}^d} \sum_{\tau=1}^{k-1} (w^T \phi(s_h^\tau, a_h^\tau) - V(s_{h+1}^\tau))^2 + \|w\|_2^2. \tag{1}$$

For the tabular case, we simply use the empirical estimation of $P_h$:

$$\widehat{P}_h^k(s'|s, a) = \begin{cases} \frac{N_h^k(s,a,s')}{N_h^k(s,a)}, & N_h^k(s, a) > 0 \\ \frac{1}{S}, & N_h^k(s, a) = 0 \end{cases} \tag{2}$$

and define $\widehat{P}_h^k V$ in the conventional manner. Here $N_h^k(s, a, s') = \sum_{\tau=1}^{k-1} \mathbb{1}\{(s_h^\tau, a_h^\tau, s_{h+1}^\tau) = (s, a, s')\}$ and $N_h^k(s, a) = \sum_{\tau=1}^{k-1} \mathbb{1}\{(s_h^\tau, a_h^\tau) = (s, a)\}$ are the numbers of visit time.

The following *optimism bonus* $\Gamma_h^k(\cdot, \cdot)$ is sufficient to guarantee optimism in standard regret minimization RL algorithms. (The choices of $\beta_{\text{tbl}}$ and $\beta_{\text{lin}}$ is specified in the appendix)

$$\Gamma_h^k(\cdot, \cdot) \leftarrow \begin{cases} \min\{\beta_{\text{lin}} \cdot (\phi(\cdot, \cdot)^T (\Lambda_h^k)^{-1} \phi(\cdot, \cdot))^{1/2}, H\}, & \text{(Linear Case)} \\ \min\{\beta_{\text{tbl}} \cdot N_h^k(\cdot, \cdot)^{-1/2}, H\}, & \text{(Tabular Case).} \end{cases} \tag{3}$$

In our setting we enlarge the optimism bonus to the following *exploration bonus*.

$$b_h^k(\cdot, \cdot) \leftarrow \begin{cases} 3\Gamma(\cdot, \cdot), & \text{(Linear Case)} \\ C \cdot \frac{H^2 S}{N_h^k(\cdot, \cdot)} + 2\Gamma_h^k(\cdot, \cdot), & \text{(Tabular Case).} \end{cases} \tag{4}$$

We then set the optimistic Q-function as

$$\overline{Q}_h^k(\cdot, \cdot) \leftarrow \Pi_{[0, H-h+1]}[\widehat{P}_h^k \overline{V}_{h+1}^k(\cdot, \cdot) + b_h^k(\cdot, \cdot)]$$

and define the exploration policy as the greedy policy with respect to $\overline{Q}_h^k$.

## 5.3 Reward Learning & Planning Phase

After the exploration phase, we run the active reward learning algorithm introduced before on the collected dataset. In the linear setting, we replace the original action with uniform random action. We then use the learned reward function to plan for a near-optimal policy. We still add optimism bonus to guarantee optimism. The whole algorithm is presented in Algorithm 3

## 5.4 Theoretical Guarantee

**Theorem 2.** *In the linear case, our algorithm can find an $\varepsilon$-optimal policy with probability at least $1 - \delta$, with at most*

$$O\left( \frac{|\mathcal{A}|^2 d^5 \dim^3(\mathcal{F}) H^4 \iota^3}{\varepsilon^2} \right), \quad \iota = \log(\frac{HSA}{\varepsilon \delta \Delta})$$

---

**Algorithm 2** UCBVI-Exploration

---

**for** $k = 1, 2, ..., K$ **do**
  $\overline{V}_{H+1}^k \leftarrow 0, \overline{Q}_{H+1}^k \leftarrow 0$
  **for** $h = H, H-1, ..., 1$ **do**
    Estimate $\widehat{P}_h^k \overline{V}_{h+1}^k(\cdot, \cdot)$ using (1) or (2)
    Set the optimism bonus $\Gamma_h^k(\cdot, \cdot)$ using (3)
    Set the exploration bonus $b_h^k(\cdot, \cdot)$ using (4).
    Set the optimistic Q-function $\overline{Q}_h^k(\cdot, \cdot) \leftarrow \Pi_{[0, H-h+1]}[\widehat{P}_h^k \overline{V}_{h+1}^k(\cdot, \cdot) + b_h^k(\cdot, \cdot)]$
    $\pi_h^k(\cdot) \leftarrow \arg\max_{a \in \mathcal{A}} \overline{Q}_h^k(\cdot, a)$
    $\overline{V}_h^k(\cdot) \leftarrow \max_{a \in \mathcal{A}} \overline{Q}_h^k(\cdot, a)$
  **end for**
  Execute policy $\pi^k = \{\pi_h^k\}_{h \in [H]}$ to induce a trajectory $s_1^k, a_1^k, ..., s_H^k, a_H^k, s_{H+1}^k$.
**end for**
**return:** Dataset $\mathcal{D} = \{(s_h^k, a_h^k)\}_{(h,k) \in [H] \times [K]}$

---

**Algorithm 3** UCBVI-Planning

---

**Input:** Dataset $\mathcal{D} = \{(s_h^k, a_h^k)\}_{(h,k) \in [H] \times [K]}$
**for** $h = 1, 2, ..., H$ **do**
  **if** Linear Case **then**
    $\widetilde{\mathcal{Z}}_h \leftarrow \{(s_h^k, \widetilde{a}_h^k)\}_{k \in [K]}$, where $\{\widetilde{a}_h^k\}_{k \in [K]}$ are sampled i.i.d. from $\mathrm{Unif}(\mathcal{A})$
    $\hat{r}_h \leftarrow$ Active Reward Learning($\widetilde{\mathcal{Z}}_h, \Delta, \delta/(2H)$).
  **else if** Tabular Case **then**
    $\mathcal{Z}_h \leftarrow \{(s_h^k, a_h^k)\}_{k \in [K]}$
    $\hat{r}_h \leftarrow$ Active Reward Learning($\mathcal{Z}_h, \Delta, \delta/(2H)$).
  **end if**
**end for**
**for** $k = 1, 2, ..., K$ **do**
  $V_{H+1}^k \leftarrow 0, Q_{H+1}^k \leftarrow 0$
  **for** $h = H, H-1, ..., 1$ **do**
    Estimate $\widehat{P}_h^k V_{h+1}^k(\cdot, \cdot)$ using (1) or (2)
    Set the optimism bonus $\Gamma_h^k(\cdot, \cdot)$ using (3)
    Set the optimistic Q-function $Q_h^k(\cdot, \cdot) \leftarrow \Pi_{[0, H-h+1]}[\hat{r}_h(\cdot, \cdot) + \widehat{P}_h^k V_{h+1}^k + \Gamma_h^k(\cdot, \cdot)]$
    $\hat{\pi}_h^k(\cdot) \leftarrow \arg\max_{a \in \mathcal{A}} Q_h^k(\cdot, a)$
    $V_h^k(\cdot) \leftarrow \max_{a \in \mathcal{A}} Q_h^k(\cdot, a)$
  **end for**
**end for**
**return:** $\hat{\pi}$ drawn uniformly from $\{\hat{\pi}^k\}_{k=1}^K$ where $\hat{\pi}^k = \{\hat{\pi}_h^k\}_{h \in [H]}$

---

*episodes. In the tabular case, our algorithm can find an $\varepsilon$-optimal policy with probability at least $1 - \delta$, with at most*

$$O\left(\frac{H^4 SA\iota}{\varepsilon^2} + \frac{H^3 S^2 A\iota^2}{\varepsilon}\right), \quad \iota = \log(\frac{HSA}{\varepsilon\delta})$$

*episodes. In both cases, the total number of queries to the reward is bounded by $\widetilde{O}(H \cdot \dim^2(\mathcal{F})/\Delta^2)$.*

**Remark 1.** *Theorem 2 readily extends to multi-task RL setting by replacing $\delta$ with $\delta/N$ and applying a union bound over all tasks, where $N$ is the number of tasks. The corresponding sample complexity bound only increase by a factor of $\mathrm{poly} \log(N)$.*

**Remark 2.** *Standard RL algorithms require to query the reward function for at least $\Omega(\frac{\max\{\dim_R, \dim_P\}^2}{\varepsilon^2})$ times, where $\dim_R$ and $\dim_P$ stand for the complexity of the reward/transition function. See, e.g., Jin et al. [2020b], Zanette et al. [2020a], Wang et al. [2020c] for the derivation of this bound. Compared to this bound, our feedback complexity bound has two merits: 1) In practice the transition function is generally more complex than the reward function, thus*

$\max\{\dim_R, \dim_P\} \gg \dim_R$; *2) Our bound is independent of $\varepsilon$ - note that $\varepsilon$ can be arbitrarily small, whereas $\Delta$ is a constant.*

**Proof Sketch**    The suboptimality of the policy $\hat{\pi}$ can be decomposed into two parts:

$$V_1^{\pi^*} - V_1^{\hat{\pi}} \le \underbrace{|V_1^{\pi^*(\hat{r})}(\hat{r}) - V_1^{\hat{\pi}}(\hat{r})|}_{(i)} + \underbrace{|V_1^{\pi^*} - V_1^{\pi^*}(\hat{r})| + |V_1^{\hat{\pi}} - V_1^{\hat{\pi}}(\hat{r})|}_{(ii)}$$

where (i) correspond to the planning error in the planning phase (Algorithm 3) and (ii) correspond to the estimation error of the reward $\hat{r}$. By standard techniques from the reward-free RL, (i) can be upper bounded by the expected summation of the exploration bonuses in the exploration phase. In order to bound (ii), we need the learned reward function to be *universally* correct, not just on the explored region. We show that the dataset collected in the exploration phase essentially *cover* the state space (tabular case) or the latent state space. Since the reward function class has bounded complexity ($\log |\mathcal{R}|$ is bounded due to the bounded covering number of $\mathcal{F}$), the reward function learned from the exploratory dataset can generalized to a distribution induced by any policy.

# 6 Offline RL with Active Reward Learning

In this section we consider the *offline RL* setting, where the dataset $\mathcal{D}$ is provided beforehand. We show that our active reward learning algorithm can still work well in this setting. In order to give meaningful result, we assume the following *compliance* property of $\mathcal{D}$ with respect to the underlying MDP. This assumption is firstly introduced in Jin et al. [2021b]. Unlike many literature for offline RL, we do not require strong coverage assumptions, e.g., concentratability [Szepesvári and Munos, 2005, Antos et al., 2008, Chen and Jiang, 2019].

**Definition 3** (Compliance). *For a dataset $\mathcal{D} = \{(s_h^k, a_h^k)\}_{(h,k) \in [H] \times [K]}$, let $\mathbb{P}_\mathcal{D}$ be the joint distribution of the data collecting process. We say $\mathcal{D}$ is compliant with the underlying MDP if $\mathbb{P}_\mathcal{D}(s_{h+1}^k = s | \{(s_h^j, a_h^j)\}_{j=1}^k, \{s_{h+1}^j\}_{j=1}^{k-1}) = P_h(s | s_h^k, a_h^k)$ holds for all $h \in [H], k \in [K], s \in \mathcal{S}$.*

## 6.1 Algorithm and Theoretical Guarantee

At the beginning of the algorithm we call the active reward learning algorithm to estimate the reward function. Inspired by Jin et al. [2021b], we estimated the optimal Q-value $Q^k$ using *pessimistic* value iteration with empirical transition and learned reward function. The policy is defined as the greedy policy with respect to $Q^k$. The full algorithm and theoretical guarantee is stated below.

---

**Algorithm 4** LCBVI-Tabular-Offline

---

    **Input:** Dataset $\mathcal{D} = \{(s_h^k, a_h^k)\}_{(h,k) \in [H] \times [K]}$
    **for** $h = 1, 2, ..., H$ **do**
        $\mathcal{Z}_h \leftarrow \{(s_h^k, a_h^k)\}_{k \in [K]}$
        $\hat{r}_h \leftarrow$ Active Reward Learning$(\mathcal{Z}_h, \Delta, \delta/(2H))$.
    **end for**
    $\widehat{V}_{H+1} \leftarrow 0$.
    **for** $h = H, H-1, ..., 1$ **do**
        $\Gamma_h(\cdot, \cdot) \leftarrow \beta'_{\text{tbl}} \cdot (N_h(\cdot, \cdot) + 1)^{-1/2}$
        $Q_h(\cdot, \cdot) \leftarrow \Pi_{[0, H-h+1]}[\hat{r}_h(\cdot, \cdot) + \widehat{\mathbb{P}}_h \widehat{V}_{h+1}(\cdot, \cdot) - 2\Gamma_h(\cdot, \cdot)]$
        $\hat{\pi}_h(\cdot) \leftarrow \arg\max_{a \in \mathcal{A}} Q_h(\cdot, a)$
        $V_h(\cdot) \leftarrow \max_{a \in \mathcal{A}} Q_h(\cdot, a)$
    **end for**
    **return:** $\hat{\pi} = \{\hat{\pi}_h\}_{h \in [H]}$

---

**Theorem 3.** *With probability at least $1 - \delta$, the sub-optimal gap of $\hat{\pi}$ is bounded by*

$$V_1^*(s_1) - V_1^{\hat{\pi}}(s_1) \le 2\left(H\sqrt{S\log(SAHK/\delta)} \cdot \mathbb{E}_{\pi^*}\left[\sum_{h=1}^H (N_h(s_h, a_h) + 1)^{-1/2}\right]\right).$$

*And the total number of queries is bounded by $\widetilde{O}(H \cdot \dim^2(\mathcal{F})/\Delta^2)$.*

The proof of Theorem 3 is deferred to the appendix.

# 7 Numerical Simulations

We run a few experiments to test the efficacy of our algorithmic framework and verify our theory. We consider a tabular MDP with linear reward. The details of the experiments are deferred to Appendix A. Here we highlight three main points derived from the experiment.

- Active learning helps to reduce feedback complexity compared to passive learning. For instance, to learn a $0.02$-optimal policy, the active learning-based algorithm only needs $\sim 70$ queries to the human teacher, while the passive learning-based algorithm requires $\sim 200$ queries. (Figure 1, left panel)

- The noise parameter $\Delta$ plays an essential role in the feedback complexity, which is consistent with our bound. For instance, with fixed number of queries, the average error of the learned policy is $0.05, 0.02, 0.005$ for $\Delta = 0.02, 0.05, 0.1$. (Figure 1, right panel)

- When $\Delta$ is relatively large (which indicates that the reward learning problem is not inherently difficult for the human teacher), we can learn an accurate policy with much fewer queries to the human teacher compared to the number of environmental steps. For instance, for $\Delta = 0.05$, to learn a $0.01$-optimal policy, our algorithm requires $\sim 2000$ environmental steps but only requires $\sim 150$ queries. (Figure 1, left panel)

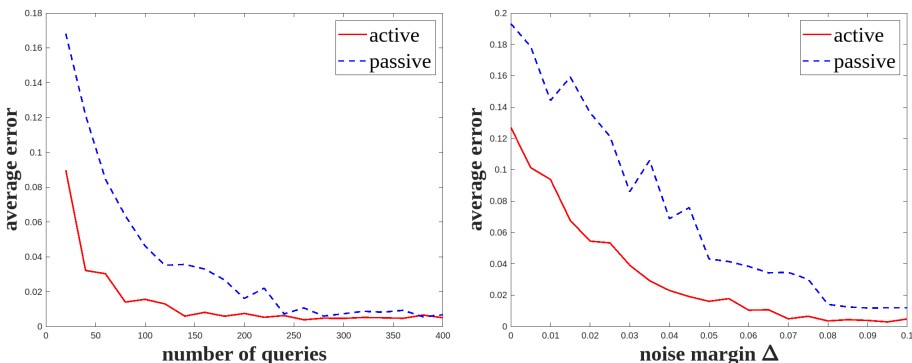

Figure 1: *Left*: average error v.s. number of queries. *Right*: the effect of the noise margin $\Delta$.

# 8 Conclusions and Discussions

In this work, we provide a provably feedback-efficient algorithmic framework that takes human-in-the-loop to specify rewards of given tasks. Our proposed framework theoretically addresses several issues of incorporating humans' feedback in RL, such as noisy, non-numerical feedback and high feedback complexity. Technically, our work integrates reward-free RL and active learning in a non-trivial way. The current framework is limited to information gain-based active learning, and an interesting future direction is incorporating different active learning methods, such as disagreement-based active learning, into our framework.

From a broad perspective, our work is a theoretical validation of recent empirical successes in HiL RL. Our results also brings new ideas to practice: it provides a new type of selection criterion that can be used in active queries; it suggests that one can use recently developed reward-free RL algorithms for unsupervised pre-training. These ideas can be combined with existing deep RL frameworks to be scalable. A limitation of the current work is that it mainly focus on theory, and we leave the empirical test of these ideas in real-world deep RL as future work.

# Acknowledgement

DK is partially supported by the elite undergraduate training program of School of Mathematical Sciences in Peking University. LY is supported in part by DARPA grant HR00112190130, NSF Award 2221871.

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
