# OpenReview forum: "Provably Feedback-Efficient Reinforcement Learning via Active Reward Learning"
_NeurIPS.cc/2022/Conference — NeurIPS 2022 Accept_

### Official Review · Reviewer_L6qb · 2022-07-10

**Rating:** 7
**Confidence:** 2
**Soundness:** 4 excellent
**Presentation:** 3 good
**Contribution:** 3 good

**Summary:**

The authors propose a theoretical framework for studying human-in-the-loop RL which splits the problem into a reward-free exploration phase and an active reward learning phase. The agent has no knowledge of the reward function and has to learn about the reward from a human expert. In the first phase the agent explores the environment using standard RFE methods. In the second phase the agent then queries the human about states it has encountered during exploration in order to learn about the reward. The authors study the sample complexity of this problem, comparing the sample complexity of exploration with the sample complexity of learning the reward. They propose an active reward learning algorithm and bound it's sample complexity as a function of the complexity of the reward function class. They then apply this algorithm to linear MDPs and offline RL deriving tighter sample complexity results for these special cases.

**Questions:**

Would it be possible to extend the proposed framework to rewards that are not binary, and noise models that are not bounded (e.g., sub-Gaussian noise)?



**Minor**

- The abstract on OpenReview has an error in the Latex syntax. Also, it should be a single paragraph.
- There are some capitalization issues and inconsistencies in the references ("bellman" -> "Bellman", "markov" -> "Markov", "q-learning" -> "Q-learning", etc.)
- The paper should probably cite other work on active reward learning such as Sadigh et al. 2017

**Limitations:**

The paper would benefit from a discussion of the assumptions made for the theoretical results, in particular, binary rewards and bounded noise. It should also discuss if the approach could be extended to nonlinear rewards.

I did not find a discussion of potential broader impact of the paper.

**Strengths And Weaknesses:**

**Strengths**
- The paper provides a nice theoretical framework to think about human-in-the-loop RL.
- The theoretical results support the empirical observation that we need much less information about the reward than interactions with the environment, which is encouraging for work on RL that relies on human feedback.
- The theoretical analysis seems strong and the results all seem sound to me (I did not check the proofs)
- The application of the theoretical analysis to offline RL is particularly nice, and supports the generality of the proposed framework.
- The paper is well-written overall and well-structured.

**Weaknesses**
- The setup makes some strong and impractical assumptions, such as assuming binary rewards and bounded noise.
- The proposed framework of separating an exploration and an active learning phase is not novel and implicitly assumed in a lot of prior work on active reward learning. Practical work often finds that iterating between exploration and active learning is beneficial, which the theoretical framework does not capture.
- The paper could be clearer about how the theoretical framework connects to practical applications.
- The paper lacks any empirical evaluation. Having at least a small empirical evaluation of the proposed method, to see if the sample complexity results match empirical performance at least in simple experiments, would make the paper much stronger in my opinion.

---

> ### Author Response · Authors · 2022-08-02
> **Author response (Part I)**
>
> We thank the reviewer for recognizing the value of our paper and providing valuable feedback. We address the concerns of the reviewer as follows.
>
> $\textbf{Q1.}$ “The paper lacks any empirical evaluation. Having at least a small empirical evaluation of the proposed method, to see if the sample complexity results match empirical performance at least in simple experiments, would make the paper much stronger in my opinion.”
>
> $\textbf{A1.}$ We kindly refer the reviewer to Section A in the appendix for the simulation results. We consider a tabular MDP with linear reward. We highlight three main points derived from the experiment.
> 1. Active learning helps to reduce feedback complexity compared to passive learning. For instance, to learn a $0.025$-optimal policy, the active learning-based algorithm only needs $\sim 100$ queries to the human teacher, while the passive learning-based algorithm requires $\sim 280$ queries. (Figure 1)
>
> 2. The noise parameter $\Delta$ plays an essential role in the feedback complexity, which is consistent with our bound. For instance, with fixed number of queries, the average error of the learned policy is $0.07, 0.02, 0.005$ for $\Delta=0.02,0.05,0.1$ (Figure 2)
>
> 3. When $\Delta$ is relatively large (which indicates that the reward learning problem is not inherently difficult for the human teacher), we can learn an accurate policy with much fewer queries to the human teacher compared to the number of environmental steps. For instance, for $\Delta=0.05$, to learn a $0.01$-optimal policy, our algorithm requires $\sim 1000$ environmental steps but only requires $\sim 200$ queries. (Figure 1)
>
>
> $\textbf{Q2.}$ “The setup makes some strong and impractical assumptions, such as assuming binary rewards and bounded noise.”
>
> $\textbf{A2.}$ We respectfully disagree that the assumptions are strong. As will be mentioned in $\textbf{A6.}$, these binary rewards model natural binary feedback from a human teacher. The bounded noise assumption represents how certain the human teacher is. With that being said, our work can also address other types of rewards (see the $\textbf{A3.}$ below).
>
> $\textbf{Q3.}$ “Would it be possible to extend the proposed framework to rewards that are not binary and noise models that are not bounded (e.g., sub-Gaussian noise)?”
>
> $\textbf{A3.}$ Our framework can generalize the bounded noise assumption to the low noise assumption (a.k.a, Tsybakov noise) [Mammen and Tsybakov, 1999], [Tsybakov, 2004], which is another standard assumption in the active learning literature. Concretely, we assume that there exists constants $\alpha\in[0,\infty)$ and $c>0$, such that for any policy $\pi$, level $h\in[H]$, and $\epsilon>0$,
>
> $P(|f^*_h (s_h,a_h)-1/2|\leq\epsilon \big| s_h, a_h \sim \pi )<c \cdot \epsilon^\alpha.$
>
> With this assumption, we can derive a similar feedback complexity bound with $\frac{1}{\Delta}$ replaced by $(\text{poly}(\frac{1}{\epsilon},d,\dim(\mathcal{F}),H,|\mathcal{A}|))^{\frac{1}{\alpha}}$ and a polynomial sample complexity bound. In this case, the difficulty of the reward learning problem depends on the exponent $\alpha$.
>
> Regarding non-binary rewards, we give an approach for generalizing binary rewards to $n$-uniform discrete rewards. We take $n=3$ for example, and the case for general $n$ is identical. In this case, the reward takes value from $\{0,\frac12, 1\}$. In each query, the human teacher chooses from $\{0, \frac12, 1\}$, which can be interpreted as “bad”, “average”, and “good” actions. We assume the probabilities of choosing $0, \frac12, 1$ are $p_0$, $p_1$, and $p_2$. The human response model satisfies:
>
> $0 * p_0 +\frac12 * p_1 + 1 * p_2 = f^*(z)$
>
> where $f^*$ belongs to a function class with bounded complexity. We assume the true reward of $z$ is determined by $f^*(z)$. Concretely, $r^*(z)=0,\frac12,1$ if $f^*(z)$ belongs to $[0\frac14),(\frac14,\frac34),(\frac34,1]$, respectively. The bounded noise assumption becomes that $f^*(z)$ can not be too near the decision boundary, i.e., $|f^*(z)-\frac14|, |f^*(z)-\frac34|>\Delta$ for all $z$.The other parts of the algorithm are similar to that with binary rewards. The sample and feedback complexity are the same as in the binary reward case.
>
> $\textbf{Q4.}$ “The proposed framework of separating an exploration and an active learning phase is not novel and implicitly assumed in a lot of prior work on active reward learning. Practical work often finds that iterating between exploration and active learning is beneficial, which the theoretical framework does not capture.”
>
> $\textbf{A4.}$ Very good point! Our current framework takes a multi-task point of view – it only explores the environment once to learn the representations and then is able to solve multiple different tasks. If there is only a single task, it might be beneficial to take an iterative approach to alternatively improve the learning in reward and environment. We leave this as future work.

---

> > ### Comment · Reviewer_L6qb · 2022-08-06
> > **Thanks for the response!**
> >
> > I appreciate the authors' clear response that addresses my open questions. I still think that assuming binary rewards and bounded noise is quite limiting. But I appreciate the authors adding a discussion of extensions to other discrete rewards and other noise models to the appendix, as well as adding simulation experiments. As these do address the key concerns I expressed about the paper, I will increase my score from 6 to 7, and I think the paper should be accepted.

---

> > > ### Author Response · Authors · 2022-08-09
> > > **Thank you!**
> > >
> > > We thank the reviewer for the positive feedback! We have uploaded a revision of our paper. In the revision,
> > >
> > > - We fixed the typos and format errors the reviewer mentioned.
> > >
> > > - We added discussions about the binary reward and bounded noise assumptions in the main paper following the reviewer's suggestion.
> > >
> > > - We added generalizations of our results to discrete reward and low noise assumptions in Appendix F, which are inspired by the reviewer's question.
> > >
> > > - We discussed the practical impacts and limitations of the current work in Section 8, following the reviewer's suggestion.
> > >
> > > - We added citations on active reward learning following the reviewer's suggestion.
> > >
> > > Besides, experiments are included in Appendix A and will be added to the main paper in the final version.
> > >
> > > We thank the reviewer again for helping us strengthen the paper; we appreciate your efforts.

---

> ### Author Response · Authors · 2022-08-02
> **Author response (Part II)**
>
> $\textbf{Q5.}$ “The paper could be clearer about how the theoretical framework connects to practical applications.”
>
> $\textbf{A5}$ Thank you for the suggestion. Here are more discussions about how the theoretical framework connects to practical applications.
>
> As you mentioned, our work mainly serves as a theoretical validation of recent empirical successes in HiL RL. At the same time, our results also bring new ideas that can be useful for HiL RL applications: our work provides a new type of selection criterion that can be used in active queries; our work suggests that one can use recently developed reward-free RL algorithms for unsupervised pre-training. These ideas can be potentially combined with existing deep HiL RL frameworks to be scalable. We will leave empirically testing these ideas in real-world deep HiL RL as a future work but will further discuss this matter in the next version of the paper.
>
> $\textbf{Q6.}$ “The paper would benefit from a discussion of the assumptions made for the theoretical results, in particular, binary rewards and bounded noise.”
>
> $\textbf{A6.}$ Thank you for the suggestion. Here are more discussions about the assumptions we are about to add in the next version.
>
> Binary rewards capture a large group of RL problems that are suitable for having human-in-the-loop. A representative group of problems is binary judgments: For example, suppose we want a robot to learn to do a backflip. A human teacher will judge whether a flip is successful and assign a reward of 1 for success and 0 for failure.
>
> Regarding the bounded noise assumption, the value of the margin $\Delta$ depends on the intrinsic difficulty of the reward learning problem and the capacity of the human teacher. For example, if the reward is rather easy to specify and the human teacher is a field expert, and can always give the right answer with a probability of at least 80%, then $\Delta$ will be 0.3. But if the learning problem is hard or the human teacher is unfamiliar with the problem and can only give near-random answers, then $\Delta$ will be very small. But in that case, we won’t hope the human teacher can help us in the first place. So a typical good value for $\Delta$ should be a constant. Besides, as mentioned above, this bound noise assumption can be replaced by the low noise assumption, which might be more practical for some problems.
>
> $\textbf{Q7.}$ “It should also discuss if the approach could be extended to nonlinear reward.”
>
> $\textbf{A7.}$ We would like to highlight that the function class $\mathcal{F}$ for the reward model is not assumed to be linear. We only assume that $\mathcal{F}$ has bounded eluder dimension and Kolmogorov dimension. There are multiple non-linear function classes that satisfy this assumption, e.g., generalized linear functions and polynomials with finite degrees.
>
> $\textbf{Q8.}$ “Format error in the abstract, capitalization issues and inconsistencies in the references, missing citations on active reward learning”
>
> $\textbf{A8.}$ Thanks for pointing these out. We will fix them in the next version.

---

### Official Review · Reviewer_MpPD · 2022-07-11

**Rating:** 7
**Confidence:** 3
**Soundness:** 3 good
**Presentation:** 2 fair
**Contribution:** 3 good

**Summary:**

This paper aims to solve the RL problem in a setting where the rewards are initially unknown. The algorithms proceeds in two stages. First, in the exploration phase, the algorithm collects samples form the environment to learn about the transitions from state to state. Then, the algorithm queries the user about rewards for specific state-action pairs and comes up with an approximation to the optimal policy. The paper shows a bound on the sub-optimality gap on the resulting policy. The analysis is restricted to the case where the true reward is binary.

Minor points:
 - in line 119, the definition of the norm of a function wrt. to the set does not use the set Z on the right hand side. You have to change the definition so that the summation is over the elements of Z.
- it may make sense to briefly contrast your approach to [1] in the Active Learning paragraph of the "Related Work" section. They solve a related reward active learning problem, although in a slightly different setting (known transition model, unknown reward function).

[1] Lindner, D., Turchetta, M., Tschiatschek, S., Ciosek, K., Krause, A. (2021) Information Directed Reward Learning for Reinforcement Learning NeurIPS 2021.

**Questions:**

1. Your analysis is restricted to RL problems where the reward is binary (1 or 0 for every state-action pair). Can you offer any intuitions about how restricted a class of problems this is? In other words, how does the set of RL problems with binary rewards compare to the set of RL problems with rewards bounded in [0,1]?
2. Can you offer any intuitions about how well your method will scale to deep RL / larger-scale benchmarks?
3. The noise margin \Delta plays a huge role when determining the required number of queries. Can you give any insights about what the values of \Delta will be for realistically large MDPs?

**Limitations:**

The authors briefly mention limitations of the approach in the conclusion, but do not discuss them at length. In my view, the biggest limitations are: (1) it is unclear how the method scales and (2) it is unclear what the noise margin will be for realistic problems (a small margin makes the required number of queries in Theorem 3 very large).

I do not see any ethical problems with this paper.

**Strengths And Weaknesses:**

Strengths:
- The problem of finding out enough information about the reward function of an MDP to be able to come up with a close-enough idea of what the optimal policy is is a major challenge and crucial to unlock many applications of RL. This paper makes a good attempt directed towards solving this problem.
- The theoretical results paint a complete picture of the performance of the agent trained using this method.

Weaknesses:
- There is no empirical evaluation in the main paper and evaluation in the supplementary material is very limited. This is not a deal-breaker since good theoretical papers can by viable contributions to NeurIPS even without experiments, but including them would make the submission stronger.
- Presentation of the material is dry and dense at times. It could be improved by providing more intuitions about what the practical significance of each result is.

---

> ### Author Response · Authors · 2022-08-02
> **Author response**
>
> We thank the reviewer for recognizing the value of our paper and providing valuable feedback. We also thank the reviewer for the writing suggestions. We address the concerns of the reviewer as follows.
>
> $\textbf{Q1.}$ “In line 119, the definition of the norm of a function wrt. to the set does not use the set Z on the right hand side. You have to change the definition so that the summation is over the elements of Z.”
>
> $\textbf{A1.}$ Thanks for pointing out this typo. We will fix it in the next version.
>
> $\textbf{Q2.}$ “It may make sense to briefly contrast your approach to [Lindner et al. 2021] in the Active Learning paragraph of the "Related Work" section.”
>
> $\textbf{A2.}$ Thanks for pointing out this paper. It is an empirical study-focused paper and is indeed very related to our work. They provided a general method called IDRL, which uses a Bayesian reward model to select queries that maximize the information gain. Besides the differences such as known transitions, they do not provide finite query complexity bounds. Hence it is not straightforward to compare with our work. Yet, we believe the techniques established in the work can be potentially applied to prove finite query bounds for their approach.
>
> $\textbf{Q3.}$ “Your analysis is restricted to RL problems with binary rewards. Can you offer any intuitions about how restricted a class of problems this is, compared to the standard RL problems with rewards bounded in [0,1]?”
>
> $\textbf{A3.}$ We believe RL problems with binary rewards represent a large group of RL problems that are suitable and relatively easy for having human-in-the-loop. A representative group of problems is the binary judgments: For example, suppose we want a robot to learn to do a backflip. A human teacher will judge whether a flip is successful and assign a reward of 1 for success and 0 for failure.
>
> Furthermore, our approach can be generalized to $n$-uniform discrete rewards. We take $n=3$ for example, and the case for general $n$ is similar. In this case, the reward takes value from $\{0,\frac12, 1\}$. In each query, the human teacher chooses from $\{0, \frac12, 1\}$, which can be interpreted as “bad”, “average”, and “good” actions. We assume the probabilities of choosing $0, \frac12, 1$ are $p_0$, $p_1$, and $p_2$. The human response model satisfies:
>
> $0 * p_0 +\frac12 * p_1 + 1 * p_2 = f^*(z)$
>
> where $f^*$ belongs to a function class with bounded complexity. We assume the true reward of $z$ is determined by $f^*(z)$. Concretely, $r^*(z)=0,\frac12,1$ if $f^*(z)$ belongs to $[0\frac14),(\frac14,\frac34),(\frac34,1]$, respectively. The bounded noise assumption becomes that $f^*(z)$ can not be too near the decision boundary, i.e., $|f^*(z)-\frac14|, |f^*(z)-\frac34|>\Delta$ for all $z$. The other parts of the algorithm are similar to that with binary rewards. The sample and feedback complexity are the same as in the binary reward case.
>
>
> $\textbf{Q4.}$ “Can you offer any intuitions about how well your method will scale to deep RL/larger-scale benchmarks?”
>
> $\textbf{A4}$ As a general theoretical framework, our work should be viewed more appropriately as a theoretical validation of recent empirical successes in deep HiL RL. The theoretical results support the empirical observation that we need much less information about the reward than interactions with the environment, which is encouraging for work on RL that relies on human feedback. (quote from reviewer L6qp)
>
> On the other hand, our results also bring new ideas that can be useful for deep HiL RL applications: our work provides a new type of selection criterion that can be used in active queries; our work suggests that one can use recently developed reward-free RL algorithms for unsupervised pre-training.
> These ideas can be potentially combined with existing DRL frameworks to be scalable. We will leave empirically testing these ideas in real-world deep RL as future work.
>
> $\textbf{Q5.}$ “The noise margin $\Delta$ plays a huge role when determining the required number of queries. Can you give any insights about what the value of $\Delta$ will be for realistically large MDPs?”
>
> $\textbf{A5.}$ The answer depends on the intrinsic difficulty of the reward learning problem and the capacity of the human teacher. For example, if the reward is rather easy to specify and the human teacher is a field expert, and can always give the right answer with a probability of at least 80%, then $\Delta$ will be 0.3. But if the learning problem is hard or the human teacher is unfamiliar with the problem and can only give near-random answers, then $\Delta$ will be very small. But in that case, we won’t hope the human teacher can help us in the first place. So a typical good value for $\Delta$ should be a constant.

---

> > ### Comment · Reviewer_MpPD · 2022-08-09
> > **Thank you for the response.**
> >
> > I wanted to thank the authors for the clarifications, particularly for the discussion of n-uniform rewards.
> > Since this addresses an issue I had with the paper, I decided to increase my score to 7.

---

### Official Review · Reviewer_oq5m · 2022-07-12

**Rating:** 6
**Confidence:** 3
**Soundness:** 3 good
**Presentation:** 3 good
**Contribution:** 3 good

**Summary:**

Summary
-------

To better understand human-in-the-loop RL, this paper aims to provide a
theoretical understanding of the feedback required from a human to learn
a reward function. The authors proposes a provable active-learning RL
algorithm, which queries a human on feedback for its actions. The
proposed algorithm needs to query the reward function for feedback
$O(H dim_R)$ times, in contrast to standard RL algorithm which require
at least $\Omega(\text{poly}(d,
\frac{1}{\epsilon}))$. This is a purely theoretical paper, and there are
no experiments.




**Questions:**


Detailed Comments
-----------------

-   Simple synthetic experiments with a programmatic reward function
    that can be queried would help elucidate the main points, and also
    enable comparisons against simple baselines.
-   The order of the number of queries needed for your algorithm depends
    on the complexity of the function class representing the reward,
    whereas the bound on \"typical\" RL algorithms dpeends on the
    complexity of the environment transition. How do these bounds
    compare for different classes of problems? I can imagine some MDPs
    with very simple transitions and complicated reward function (and
    vice-versa). How should these differences be reconciled?

Minor Comments
--------------

-   Line 122: Mistakenly refers to three phases, when it seems there are
    only two.

**Limitations:**

The conclusion section, which the authors point to, only discusses future work and not the limitations of the algorithm nor negative societal impact. As a theory paper, societal impact is difficult to ascertain. Limitations in the theoretical analysis are somewhat discussed throughout the paper, but it would be nice to see candid discussion in the conclusion on the overall limitations of this approach.

**Strengths And Weaknesses:**


Decision
--------

Overall, I think this paper is interesting in the problem that it
studies and the theoretical results seem important. I am not an expert
in this area, however, and will tentatively rate the paper at slightly
below the acceptance threshold. The submission is held back by some open
questions, namely the comparison of the bound derived versus the bound
of standard RL algorithms. Experiments in a toy environment, with an
unknown but programmatic reward function that can be queried, would also
be nice and strengthen confidence in the importance of the derived
algorithm.

Strengths
---------

-   The problem studied, that of optimally querying a human on feedback
    for reward, is an interesting line of study that deserves a
    theoretical analysis.
-   The bound derived is surprisingly elegant, and the presentation of
    the theory is mostly readable.

Weaknesses
----------

-   Experiments in a toy environment, with some programmatic reward
    function that can be queried, would help contextualize the proposed
    algorithm and the tightness of the bound.
-   The biggest open question is how to contextualize the bound derived,
    which depends on a measure of complexity of the reward function,
    with the standard bound that depends on the complexity of of the
    transition.

---

> ### Author Response · Authors · 2022-08-02
> **Author Response**
>
> We thank the reviewer for recognizing the value of our work and providing valuable feedback. We address the concerns of the reviewer as follows.
>
> $\textbf{Q1.}$ “Experiments in a toy environment, with some programmatic reward function that can be queried, would help contextualize the proposed algorithm and the tightness of the bound.”
>
> $\textbf{A1.}$ We kindly refer the reviewer to Section A in the appendix for the simulation results. We consider a tabular MDP with linear reward. We highlight three main points derived from the experiment.
> 1. Active learning helps to reduce feedback complexity compared to passive learning. For instance, to learn a $0.025$-optimal policy, the active learning-based algorithm only needs $\sim 100$ queries to the human teacher, while the passive learning-based algorithm requires $\sim 280$ queries. (Figure 1)
>
> 2. The noise parameter $\Delta$ plays an essential role in the feedback complexity, which is consistent with our bound. For instance, with fixed number of queries, the average error of the learned policy is $0.07, 0.02, 0.005$ for $\Delta=0.02,0.05,0.1$ (Figure 2)
>
> 3. When $\Delta$ is relatively large (which indicates that the reward learning problem is not inherently difficult for the human teacher), we can learn an accurate policy with much fewer queries to the human teacher compared to the number of environmental steps. For instance, for $\Delta=0.05$, to learn a $0.01$-optimal policy, our algorithm requires $\sim 1000$ environmental steps but only requires $\sim 200$ queries. (Figure 1)
>
> $\textbf{Q2.}$ “The order of the number of queries needed for your algorithm depends on the complexity of the function class representing the reward, whereas the bound on "typical" RL algorithms depends on the complexity of the environment transition. How do these bounds compare for different classes of problems? I can imagine some MDPs with very simple transitions and complicated reward function (and vice-versa). How should these differences be reconciled?”
>
> $\textbf{A2.}$ That’s a good point. First, we believe that in practice, the transition function is generally more complex than the reward function. For example, for many games like Go or Dota2, the reward function only depends on some features of the image, while the transition function depends on the whole image. Also, the reward can be sparse for many applications, like some control problems.
>
> Second, even when the reward function is more complex than the transition function, our bound is still better than the bound derived from standard RL algorithms. The latter actually scales at least with ${\Omega}(\frac{d^2}{\epsilon^2})$, where $d=\max ( \dim_R, \dim_P ) $ ($\dim_R$ and $\dim_P$ stand for the complexity of the reward/transition function). When $\dim_R > \dim_P$, the $d^2$ term matches our bound while the dependence on $\epsilon$ makes it much worse. Note that $\epsilon$ can be made arbitrarily small, whereas $\Delta$ is a constant.
>
> $\textbf{Q3.}$ “Line 122: Mistakenly refers to three phases, when it seems there are only two.”
>
> $\textbf{A3.}$ Thanks for pointing this out. We will fix it.

---

> > ### Comment · Reviewer_oq5m · 2022-08-08
> > **Thank you for the clarification.**
> >
> > Regarding Q1: I did not realize that the experiments were in the appendix. Thank you for pointing this out and including the appendix in the main submission PDF. As reviewer MpPD points out, the experimental evaluation is limited but this is not a shortcoming as the results demonstrate feedback efficiency and relative robustness to a range of possible \Delta.
> >
> > Regarding Q2: thank you for clarifying this point as well. Indeed, my main concern was the relative complexity of the state and reward function. While the reward function is generally less complex than the state transition, it was unclear that the latter bound depended on the maximum between the two. The dependence on \Delta, instead of \epsilon, in your bound is a result of assumption 2. Is this because the previous bound does not make use of this assumption? Also, in an attempt to clarify this detail for myself, it is not clear which paper the poly(d, 1/\epsilon) is shown for reward-querying. A reference to this in your submission would be good for the uninitiated reader.
> >
> > After reviewing the changes in the most recent submission, as well as the discussion and other reviews, I am comfortable increasing my score to a 6.

---

> > > ### Author Response · Authors · 2022-08-09
> > > **Thank you for your reply!**
> > >
> > > Dear Reviewer oq5m:
> > >
> > > Thank you for your reply and for raising the new questions. Please refer to the answers below.
> > >
> > > $\textbf{Q1.}$ “Also, in an attempt to clarify this detail for myself, it is not clear which paper the poly(d, 1/\epsilon) is shown for reward-querying. A reference to this in your submission would be good for the uninitiated reader.”
> > >
> > > $\textbf{A1.}$ Standard RL algorithms require to query the reward function in each time step. Thus the $\operatorname{poly}(d,1/\epsilon)$ bound corresponds to the sample complexity bound, i.e., the total number of interactions with the environment. The literature has tended to achieve low regret for RL with linear/general function approximation. However, any low-regret algorithm can be used to obtain near-optimal policy using the online-to-batch conversion (see, e.g., Section 3.1 of [Jin et al., 2018]).
> > >
> > > With the online-to-batch conversion, the current SOTA result for linear MDP is [Zanette et al., 2020], which achieves a sample complexity bound of $\tilde{O}(d^2\cdot \operatorname{poly}(H)/\epsilon^2)$, where $d$ is the dimension of the feature extractor in the linear MDP.  But the algorithm in [Zanette et al., 2020] is computationally inefficient, and computationally efficient algorithms (e.g., [Jin et al., 2020]) at best achieve $\tilde{O}(d^3\cdot \operatorname{poly}(H)/\epsilon^2)$. For RL with general function approximation, [Wang et al., 2020] design a computationally efficient algorithm with sample complexity bound $\tilde{O}(d^4\cdot \operatorname{poly}(H)/\epsilon^2)$, where $d$ depends on the eluder dimension of the function class $\mathcal{F}$ used to learn the optimal Q-function.
> > >
> > > Thank you for the suggestion; we will add these references to the final version.
> > >
> > > $\textbf{Q2.}$ “Indeed, my main concern was the relative complexity of the state and reward function. While the reward function is generally less complex than the state transition, it was unclear that the latter bound depended on the maximum between the two.”
> > >
> > > $\textbf{A2.}$ In linear MDP [Yang and Wang, 2019][Jin et al., 2020][Zanette et al., 2020], it is assumed that both the reward and the transition functions are linear in the feature extractor. Thus the dimension of the feature extractor depends on the maximum complexity of the reward/transition function.
> > >
> > > In works on (computationally efficient) algorithms for RL with general function approximation, the following closedness assumption is usually assumed (see, e.g., [Wang et al., 2020], [Feng et al., 2021], [Ishfaq et al., 2021]). Here $\mathcal{F}$ is the function class used to learn the optimal Q-function.
> > >
> > >
> > > $\textbf{Assumption.}$(closedness) For any $h\in[H]$ and $V:\mathcal{S}\rightarrow [0,H]$,  there exists $f_V \in \mathcal{F}$ such that for all $(s,a)\in \mathcal{S}\times\mathcal{A}$,
> > >
> > > $f_V(s,a)=r_h(s,a)+\sum_{s'\in \mathcal{S}}P_h(s'|s,a)V(s').$
> > >
> > > Note that if the reward function is more complex than the transition function, the closedness assumption would require the function class $\mathcal{F}$ to at least cover the reward function space. Hence, the sample complexity bounds in these works depend on the eluder dimension of $\mathcal{F}$, thus the maximum complexity of reward/transition function (up to $\operatorname{poly}(H)$ factors).
> > >
> > > $\textbf{Q3.}$ “The dependence on \Delta, instead of \epsilon, in your bound is a result of assumption 2. Is this because the previous bound does not make use of this assumption?“
> > >
> > > $\textbf{A3.}$ The dependence on $\Delta$ instead of $\epsilon$ is a result of our active learning algorithm. A major purpose of active learning is to get bounds with better scaling in $\epsilon$ (see [Hanneke et al., 2014] and the bounds derived there). The final scaling in the bound depends on the assumption - if we replace Assumption 2 with the low noise assumption, the feedback complexity bound will have a (mild) dependency on $\epsilon$ (See Appendix F.3 in our paper for details).
> > >
> > > Reference:
> > >
> > > [Feng et al., 2021] Provably Correct Optimization and Exploration with Non-linear Policies (ICML 2021)
> > >
> > > [Hanneke et al., 2014] Theory of disagreement-based active learning
> > >
> > > [Jin et al., 2018] Is Q-learning Provably Efficient? (NeurIPS 2018)
> > >
> > > [Jin et al., 2020] Provably Efficient Reinforcement Learning with Linear Function Approximation (COLT 2020)
> > >
> > > [Ishfaq et al., 2021] Randomized Exploration for Reinforcement Learning with General Value Function Approximation (ICML 2021)
> > >
> > > [Wang et al., 2020] Reinforcement Learning with General Value Function Approximation: Provably Efficient Approach via Bounded Eluder Dimension (NeurIPS 2020)
> > >
> > > [Yang and Wang 2019] Sample-Optimal Parametric Q-Learning Using Linearly Additive Features (ICML 2019)
> > >
> > > [Zanette et al., 2020] Learning Near Optimal Policies with Low Inherent Bellman Error (ICML 2020)

---

> > > ### Author Response · Authors · 2022-08-09
> > > **Revision Uploaded**
> > >
> > > Dear Reviewer oq5m:
> > >
> > > Thank you again for your suggestion. We have uploaded a revision of our paper. In the revision, we rewrote Remark 2 and added the necessary references for deriving the $\Omega(\operatorname{poly}(d,1/\epsilon))$ bound.

---

> ### Author Response · Authors · 2022-08-08
> **Looking forward to your feedback.**
>
> Dear Reviewer oq5m:
>
> Thanks again for your valuable feedback, and we are wondering whether our response addresses your concerns (especially the open question you raised). Following your suggestions,
>
> - We compared our feedback complexity bound with the bound derived from standard RL algorithms. We showed that our bound is still better even when the reward function is more complex than the transition function.
>
> - We provided experiments to elucidate and validate the main points made in the paper.
>
> - We discussed the limitations of the current work in the conclusion section.
>
> We also revised our paper accordingly. Since the author-reviewer discussion period is closing soon, we are looking forward to your further comments on our response. We are happy to answer any questions or concerns that remain.

---

### Official Review · Reviewer_GgBc · 2022-07-13

**Rating:** 6
**Confidence:** 4
**Soundness:** 3 good
**Presentation:** 3 good
**Contribution:** 3 good

**Summary:**

This paper studies the problem of active reward learning in reinforcement learning. The objective is to design an algorithm that  allows the user, after limited access to a few reward queries, to not only reconstruct the reward function but also use it to produce an $\epsilon$-optimal policy. The authors assume the rewards are in $\{ 0,1\}$ and that the human feedback is random such that for any state action pair and episode step  $s,a,h$  the observed human feedback $Y$ satisfies,

$$\mathbb{P}(Y | s,a,h) = f_h^*(s,a,h)$$

Such that $f_h^* \in \mathcal{F}$ for some known function class $\mathcal{F}$. It is further assumed that the true rewards are a thresholded version of the human feedback so that the reward of $s,a,h$ equals $1$ if $f_h^*(s,a,h) > 1/2$ and $0$ otherwise. They make use of a Massart noise assumption to ensure the learnability of this threshold function. It is possible to show that sufficient samples from the human labeler are enough to learn the thresholded reward even over all state actions in a data pool. The number of labels needed depends only on the gap $\Delta$ (from the Massart noise assumption), and the complexity of the function class $\mathcal{F}$. In short even if the dataset is large, a dataset size independent number of samples are needed to learn the reward function over all of it.

The authors then combine this result with a reward free exploration scheme to first collect sufficient data from either a tabular or a linear MDP and use it for the purpose of reward labeling. The reward labeling results described above imply that in order to learn the reward function over all the datapoints in the data batch generated by free exploration it is enough to query the rewards over a number of state action pairs only dependent on the noise condition and the size of the function class.

They complement their discussion with some commentary on how to extend their results to the setting of offline RL.


**Questions:**

I think there should be some citations in the reward free section that are missing. For example "Reward-Free Model-Based Reinforcement Learning with Linear Function Approximation" by Weitong Zhang, Dongruo Zhou, Quanquan Gu discusses the reward free setting for linear MDPs.

Qs
a) The margin $\Delta$ is assumed to be known. What happens if this quantity is unknown? Have the authors thought how to bypass this issue?
b) In Algorithm 1 the underlying true reward is produced by thresholding $\hat{f}$. This is a member of the covering. Is there a reason why $\tilde{F}$ is not enough? This could be better explained in the main. [ I didn't look at the appendix]


**Limitations:**

This work poses no negative societal impact.

**Strengths And Weaknesses:**

1) Strengths. The setting studied by the paper seems novel, and the discussion that these results may bring about would represent a welcome addition to the reinforcement learning literature. Developing theoretical understanding of HiL algorithms and how it is that in practice these algorithms do not require massive amounts of data is very important. Thus, the assumptions used in this work are not to be thought as limiting but instead as ways of developing understanding of these issues.

2) Weaknesses. The technical contribution of this work is limited. The algorithms presented and their analysis in my understanding is mainly lifted from the existing literature. The Active Reward Learning guarantees are not really novel or surprising. The reward free section lacks citations but these results are already present in the literature.

Despite 2) I think the paper has merit because of the introduction of this setup.

---

> ### Author Response · Authors · 2022-08-02
> **Author Response**
>
> We thank the reviewer for recognizing our work's value and providing an excellent summary of our results. Now we address some of the concerns and questions raised by the reviewer.
>
> $\textbf{Q1.}$ “The technical contribution of this work is limited. The algorithms presented and their analysis in my understanding is mainly lifted from the existing literature. The Active Reward Learning guarantees are not really novel or surprising. The reward free section lacks citations but these results are already present in the literature.”
>
> $\textbf{A1.}$ Indeed, our algorithms and analysis are based on reward-free learning and active learning. However, we would like to highlight the technical contributions that the reviewer might not be fully aware of: (1) we don’t have full access to the reward and are required to query the reward function for only a small number of places. In contrast, existing reward-free/reward-agnostic algorithms assume full access to the true reward function. (2) Existing active learning algorithms/analyses require access to a fixed distribution that covers the support of the reward. However, in the RL setting, the data distribution can change and may not cover the support of the reward function. As a result, either of the existing approaches does not solve our problem. By carefully leveraging the best of both worlds, our approaches can guarantee sufficient coverage to the state-action space with a reward-free exploration phase; and only query the unknown reward function at a few places with an active learning phase. Furthermore, to ensure the analysis work smoothly, we derive a novel analysis that utilizes the latent structure of the MDP – to the best of our knowledge, we are not aware of a similar analysis in the literature.
>
> $\textbf{Q2.}$ “I think there should be some citations in the reward free section that are missing. ”
>
> $\textbf{A2.}$ Thanks for pointing out the citations. We will make sure to add all of them in the next version.
>
> $\textbf{Q3.}$ “What happens if the margin $\Delta$ is unknown?”
>
> $\textbf{A3.}$ Good point. One potential approach would be using a binary search for guessing the value of $\Delta$ – this only introduces a log factor to the asymptotic sample complexity as we only need to guess logarithmically many times. For instance, we set $\Delta’=1/(2^n), (n=1,2,...)$ and run Algorithm 2 and Algorithm 3 repeatedly.  For each $\Delta’$, after learning the human model $\tilde{f}$ in Algorithm 1, we test whether for each data point $z_i$ in the data pool we have $\tilde{f}(z_i)>\Delta'/2$. If this is true, we continue the algorithm and output a near-optimal policy as usual. Otherwise, we halt the algorithm, replace $\Delta’$ with $\Delta’/2$, and rerun the whole algorithm. The doubling schedule implies that the smallest guess is at least $\Delta/2$. Besides, we also need to adjust the confidence parameter to $\delta/(n(n+1))$, which does not change the asymptotic sample complexity. The solution for the offline setting is identical. We will add a formal theorem in the next version.
>
> $\textbf{Q4.}$“Why do we need to round $\tilde{f}$ to the covering?”
>
> $\textbf{A4.}$ We round $\tilde{f}$ to ensure there are a finite number of possibilities of such functions – this gives us the convenience of applying union bound in our analysis. Indeed, we believe a more refined analysis would remove the requirement of rounding but will make the analysis much more involved. We will discuss this in more detail in the next version. (This is also explained in line 274 - line 275 of the main paper.)

---

> > ### Comment · Reviewer_GgBc · 2022-08-09
> > **Thanks for your response**
> >
> > “The technical contribution of this work is limited" I don't think this is a terrible situation. The setting is novel and it would be great to have more activity in this area. I appreciate the authors' answer but I still think that in terms of the techniques, the results aren't very surprising. I want to reiterate this does not make the work unacceptable for neurips publication but it does make it ineligible in my opinion for a higher score.
> >
> > "Unknown margin" Maybe this could be solved using a model selection algorithm?
> >
> > Thanks a lot for your answer. I will keep my score. I think this work deserves a serious shot at publication.

---

> > > ### Author Response · Authors · 2022-08-09
> > > **Thank you for your reply!**
> > >
> > > Dear Reviewer GgBc,
> > >
> > > We thank the reviewer for the valuable feedback and for clarifying the comments on technical contribution. Please refer to the response on the unknown margin problem below.
> > >
> > > $\textbf{Q1.}$ “"Unknown margin" Maybe this could be solved using a model selection algorithm?”
> > >
> > > $\textbf{A1.}$ As shown in Appendix F.1, we can use a binary search method to guess the value of $\Delta$, which will not harm the sample and feedback complexity. As far as we are concerned, this problem is not directly related to model selection since $\Delta$ is not a hyperparameter in our model. However, we thank the reviewer for pointing out the model selection algorithm, which can be helpful when implementing our algorithms in practice. Indeed, suppose we are not sure about the complexity of the reward function. In that case, we may use model selection algorithms to choose the hyperparameters in the function class $\mathcal{F}$, e.g., the number of features for a linear function class.

---

> > > > ### Comment · Reviewer_GgBc · 2022-08-09
> > > > **Thanks**
> > > >
> > > > Sounds good! I will keep my score.

---

### Author Response · Authors · 2022-08-07
**Paper Revision**

We thank all the reviewers for their helpful feedback. We have uploaded a revision of our submission to incorporate reviewers’ suggestions. All changes in the main paper are marked in red for clarity.
We summarize the main changes in the revision:
- We fixed typos and format errors pointed out by the reviewers.
- We added discussions about the binary reward and bounded noise assumptions in the main paper, as suggested by Reviewer MpPD and Reviewer L6qb.
- We generalized our results to discrete reward and low noise assumptions in Appendix F, as suggested by Reviewer MpPD and Reviewer L6qb.
- We added a comparison of the bounds in Remark 2, as suggested by Reviewer oq5m.
- We introduced a binary search approach to address the case where the noise margin $\Delta$ is unknown in Appendix F, as suggested by Reviewer GgBc.
- We explained the reason for rounding $\tilde{f}$ to the cover in Section 5.1, as suggested by Reviewer GgBc.
- We discussed the practical impacts and limitations of the current work in Section 8, as suggested by all the reviewers.
- We revised the related work section and added citations on reward-free RL and active reward learning in Appendix E, as suggested by all the reviewers. We will make sure to move the contents in Appendix E to Section 2 in the final version.

Besides, experiments are already included in Appendix A in the original version but not mentioned in the main paper. We will include the experiments in the main paper in the final version.

---

### Meta-Review · Area_Chair_bPgf · 2022-08-26

**Recommendation:** Accept
**Confidence:** Certain

**Metareview:**

This paper investigates human-in-the-loop RL. The framework and proposed algorithm allows an agent to reconstruct the reward function and produce a near-optimal policy after limited access to a few reward queries. The primary contributions of the work are the problem formulation, algorithm and formal results.

All reviewers agreed on acceptance. Most importantly, there was consensus that problem setting is relevant and interesting and the math is correct. The reviewers noted the techniques used are not new and the results not surprising but the formulation is novel. This is not necessarily a bad thing. There was some discussion on some of the assumptions required (binary feedback and bounded noise). In the end there was clear consensus that the paper adds a much needed theoretical framework to HIL-RL and should inspire further algorithmic work.

Things to address for camera ready:
- all the reviewers thought it was a bad idea to have related work in the appendix. The AC agrees
- the experiments in the appendix are easy to miss; more clearly reference them in the main text
- the text is not great in places; especially in the additions made to the paper in response to the reviewers.

**Award:**

No

---

### Decision · Program_Chairs · 2022-09-14

Accept